# Unified bursting strategies in ectopic and endogenous *even-skipped* expression patterns

Augusto Berrocal[1†], Nicholas C Lammers[2‡], Hernan G Garcia[1,2,3,4,5]*, Michael B Eisen[1,2,4,6]*

[1]Department of Molecular & Cell Biology, University of California at Berkeley, Berkeley, United States; [2]Biophysics Graduate Group, University of California at Berkeley, Berkeley, United States; [3]Department of Physics, University of California at Berkeley, Berkeley, United States; [4]California Institute for Quantitative Biosciences (QB3), University of California at Berkeley, Berkeley, United States; [5]Chan Zuckerberg Biohub–San Francisco, San Francisco, United States; [6]Howard Hughes Medical Institute, University of California at Berkeley, Berkeley, United States

**\*For correspondence:**
hggarcia@berkeley.edu (HGG);
mbeisen@gmail.com (MBE)

**Present address:** [†]Department of Pharmaceutical Chemistry, University of California at San Francisco, United States; [‡]Department of Genome Sciences, University of Washington, Seattle, United States

## eLife assessment

This manuscript is an **important** contribution toward understanding the mechanisms of transcriptional bursting. The evidence is considered **solid**. Questions regarding the broader advance, details of the analysis, and the models used in the analysis were addressed by the authors.

**Abstract** Transcription often occurs in bursts as gene promoters switch stochastically between active and inactive states. Enhancers can dictate transcriptional activity in animal development through the modulation of burst frequency, duration, or amplitude. Previous studies observed that different enhancers can achieve a wide range of transcriptional outputs through the same strategies of bursting control. For example, in Berrocal et al., 2020, we showed that despite responding to different transcription factors, all *even-skipped* enhancers increase transcription by upregulating burst frequency and amplitude while burst duration remains largely constant. These shared bursting strategies suggest that a unified molecular mechanism constraints how enhancers modulate transcriptional output. Alternatively, different enhancers could have converged on the same bursting control strategy because of natural selection favoring one of these particular strategies. To distinguish between these two scenarios, we compared transcriptional bursting between endogenous and ectopic gene expression patterns. Because enhancers act under different regulatory inputs in ectopic patterns, dissimilar bursting control strategies between endogenous and ectopic patterns would suggest that enhancers adapted their bursting strategies to their *trans*-regulatory environment. Here, we generated ectopic *even-skipped* transcription patterns in fruit fly embryos and discovered that bursting strategies remain consistent in endogenous and ectopic *even-skipped* expression. These results provide evidence for a unified molecular mechanism shaping *even-skipped* bursting strategies and serve as a starting point to uncover the realm of strategies employed by other enhancers.

DOI: https://doi.org/10.7554/eLife.88671

## Introduction

In animal development, enhancers, *cis*-regulatory elements that can act at a distance to modulate the transcription of genes (*Banerji et al., 1981*; *Banerji et al., 1983*; *Gillies et al., 1983*) orchestrate the formation of gene expression patterns that dictate animal body plans (*Davidson, 2010*; *Roberta, 2015*; *Lewis, 1978*). At the single-cell level, transcription of most genes has been shown to occur in stochastic pulses, or bursts, of mRNA synthesis (*Dar et al., 2012*; *Golding et al., 2005*; *McKnight and Miller, 1979*; *Raj et al., 2006*; *Senecal et al., 2014*; *Skupsky et al., 2010*; *Zenklusen et al., 2008*), and patterned developmental genes are no exception (*Berrocal et al., 2020*; *Bothma et al., 2014*; *Fukaya et al., 2016*; *Lammers et al., 2020*; *Zoller et al., 2018*). Enhancers typically feature binding sites for several transcription factors proteins. Through these binding sites, enhancers can read out transcription factor concentration and modulate transcriptional bursting dynamics of the genes they regulate (*Bothma et al., 2014*; *Bothma et al., 2015*; *Chen et al., 2018*; *Fukaya et al., 2016*; *Small et al., 1992*; *Yuh et al., 1994*).

Transcriptional bursting can be described by the two-state model of promoter activity (*Lionnet and Singer, 2012*; *Peccoud and Ycart, 1995*; *Sanchez et al., 2013*) that depicts bursts as the result of a gene promoter that switches stochastically between an inactive state, OFF, and an active state, ON, at a rate $k_{on}$. When the promoter is in its ON state, it loads RNA Pol II molecules onto the gene at a rate $r$ until, eventually, the promoter transitions back to the OFF state at a rate $k_{off}$ and mRNA synthesis stops (*Figure 1A and B*). In this model, there are multiple distinct ways that enhancers could modulate the rate of mRNA production by tuning transcriptional parameters. For instance, enhancers could upregulate transcription through an increase in burst frequency ($k_{on}$, also defined as a decrease in the interval between bursts or $k_{on}^{-1}$), burst duration ($k_{off}^{-1}$) or burst amplitude ($r$), or any combination thereof. Recently, quantitative studies have revealed striking similarities in how disparate enhancers modulate these burst parameters to control gene expression. For example, using live-imaging and statistical modeling, we previously showed that the five enhancers that form the seven stripes of *even-skipped (eve)* expression in *Drosophila melanogaster*, despite each interacting with a different set of transcription factors, employ the same kinetic strategy to control the rate of mRNA synthesis: they modulate burst frequency and amplitude, while leaving burst duration largely unchanged (*Berrocal et al., 2020*). Similarly, another study employing single-molecule mRNA FISH suggested that the transcriptional control of various *D. melanogaster* gap genes is characterized by the shared modulation of burst frequency and duration, while burst amplitude remains constant (*Zoller et al., 2018*). These two examples suggest a surprising degree of unity—but also of diversity—in the way different enhancers interact with promoters to control transcriptional bursting.

Apparent regulatory unity between various enhancers could be the result of evolutionary adaptation of enhancers to the *trans*-regulatory inputs that they experience in their endogenous regions of activity. Under this model, we would expect to observe unified bursting strategies at endogenous regions of enhancer activity, while enhancers exposed to non-endogenous regulatory inputs could exhibit different bursting strategies than those observed within their canonical domains of activity. Alternatively, unified strategies of bursting control could result from constraints determined by the biochemistry of the transcriptional processes at enhancers and promoters. In this model, enhancers would control the same set of bursting parameters regardless of the identity and concentration of the input transcription factors at concentrations that enhancers have not encountered during their evolution.

To probe these two models in the context of *D. melanogaster* development, we used the *eve* gene as a case study. Our previous work (*Berrocal et al., 2020*) only examined bursting control strategies in endogenous *eve* stripes, whose expression dynamics are, in principle, subject to evolutionary selection. To examine expression dynamics in a region presumably devoid of such evolutionary selection, in this study we induced the formation of ectopic *eve* expression patterns. Specifically, we disrupted two *eve* enhancers to expand the transcriptional activity of the *eve* gene onto ectopic regions where enhancers dictate transcriptional bursting in the presence of combinations and concentrations of input transcription factors that *D. melanogaster eve* enhancers have not encountered in their evolution. We compared bursting parameters in endogenous (*Figure 1C*) and ectopic regions of *eve* expression (*Figure 1D*) and determined that, despite endogenous regions having a higher mean transcriptional output than ectopic regions of *eve* expression, nuclei in endogenous and ectopic regions modulate their transcriptional output through the same bursting strategies: a concerted increase in promoter $k_{on}$

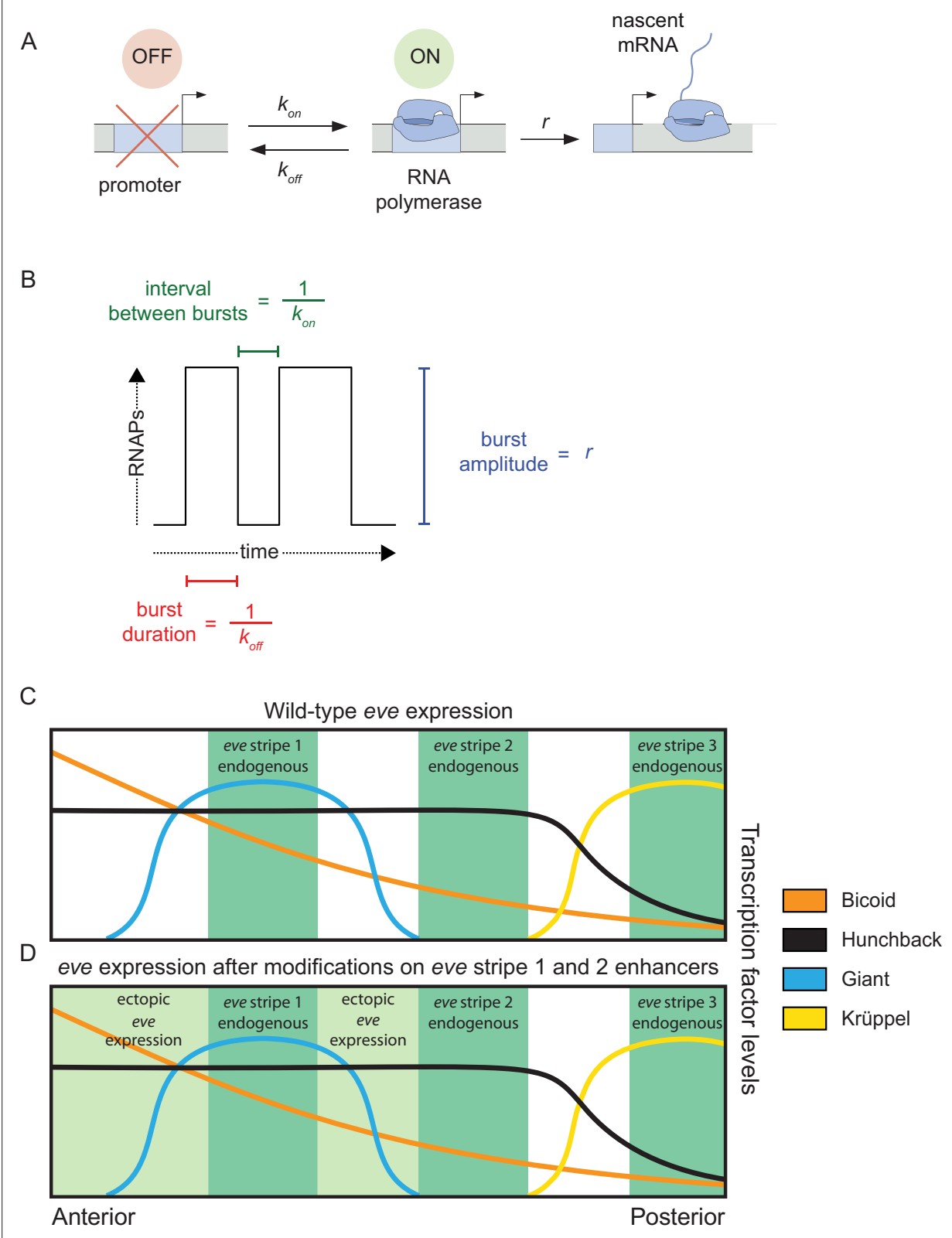

**Figure 1.** Promoter activity in endogenous and ectopic regions of *eve* expression. (**A**) According to the two-state model of promoter activity a gene promoter switches from the OFF (inactive) state to the ON (active) state at a rate $k_{on}$. When ON, the promoter loads RNA Pol II molecules and synthesizes mRNA at a rate $r$. The promoter stochastically switches back to the OFF state at a rate $k_{off}$. (**B**) The $k_{on}$, $k_{off}$, and $r$ parameters define the average interval between bursts, average burst duration, and average burst amplitude, respectively. (**C**) *eve* stripes result from the interplay of various

*Figure 1 continued on next page*

*Figure 1 continued*

activators and repressors, for instance, wild-type *eve* stripe 2 is expressed through the interplay of the activators Bicoid and Hunchback with the repressors Giant and Krüppel. The latter define the anterior and posterior boundaries of *eve* stripe 2, respectively. (**D**) Here, we coupled the disruption of the *eve* stripe 1 enhancer with the disruption of the anterior repression of *eve* stripe 2 exerted by the gap repressor Giant to drive ectopic *eve* expression anteriorly and compare bursting parameters between endogenous and ectopic expression patterns. (**C and D**) are based on ***Levine, 2013*** and ***Peel et al., 2005***.

and *r*, while $k_{off}$ remains largely unchanged. Our results suggest that *eve* enhancers have not adapted to yield particular bursting parameters within *eve* stripes and add further evidence for a unified molecular mechanism behind the modulation of *eve* transcriptional output. Our work serves as a starting point for uncovering the realm of possible bursting strategies employed by enhancers and opens new research avenues to investigate how these strategies are established at the molecular level.

## Results
### Mutating *eve* enhancers to generate ectopic expression patterns

We sought to determine whether *eve* enhancers regulate transcription by modulating the same set of bursting parameters in endogenous and ectopic *eve* expression regions. Specifically, we aimed to compare how *eve* enhancers drive transcriptional bursting in and out of the well-known seven endogenous *eve* stripes (***Frasch and Levine, 1987***; ***Hare et al., 2008***).

As our starting point, we took a previously established BAC-based *eve*-MS2 reporter system (***Berrocal et al., 2020***) that carries an ~20 kb DNA fragment around the *D. melanogaster eve* coding region containing the five *eve* enhancers responsible for regulating the expression of the seven *eve* stripes, other *cis*-regulatory elements such as neuronal and muscular regulatory elements (***Fujioka et al., 1999***; ***Fujioka et al., 2013***) that might influence *eve* stripe expression in early development (***Fujioka et al., 1999***; ***Fujioka et al., 2013***), and the late element (LE) that upregulates *eve* expression in all stripes in response to the EVE protein (***Fujioka et al., 1996***; ***Jiang et al., 1991***; ***Figure 2A***). We will refer to this construct as eveMS2-BAC (see SI section: DNA constructs and fly lines in Materials and methods). The MS2 reporter system fluorescently labels nascent mRNA molecules resulting in sites of nascent transcription appearing as puncta whose fluorescence is proportional to the number of active RNA Pol II molecules. As a result, the system allows for the visualization of transcriptional bursting at single locus resolution, in real-time, in living embryos (***Chubb et al., 2006***; ***Ferguson and Larson, 2013***; ***Garcia et al., 2013***; ***Golding et al., 2005***; ***Golding and Cox, 2004***). When inserted into the *D. melanogaster* genome, eveMS2-BAC expresses in seven stripes that recapitulate the wild-type expression of *eve* (***Figure 2B***; ***Berrocal et al., 2020***) as observed by FISH and live-imaging experiments (***Lammers et al., 2020***; ***Lim et al., 2018***; ***Luengo Hendriks et al., 2006***).

To establish an ectopic *eve* expression pattern, we modified the *eve* reporter locus (***Figure 2A***; ***Berrocal et al., 2020***). Specifically, we aimed to create an anterior expansion of *eve* stripe 2 beyond its endogenous expression domain and into ectopic regions where we could study transcriptional bursting under inputs foreign to an *eve* enhancer, for example higher levels of the activator Bicoid and the repressor Giant (Gt) (***Figure 1D***). To make this possible, we leveraged the fact that the anterior boundary of *eve* stripe 2 is established through repression by Giant (***Small et al., 1992***). Classic work by Small et al. identified a minimal regulatory element of the *eve* stripe 2 enhancer (eveS2-MRE; ***Figure 2A***) and found that deleting three Giant binding sites within this minimal enhancer produced a strong anterior expansion of *eve* stripe 2 in the context of a reporter driven by eveS2-MRE (***Small et al., 1992***).

We generated an eveMS2-BAC variant, where the three binding sites for Giant identified in the eveS2-MRE were disrupted on the complete *eve* stripe 2 enhancer (eveS1wt-eveS2Gt⁻; ***Figure 2A and C***). Live imaging experiments on eveS1wt-eveS2Gt⁻ embryos showed only transient ectopic expression at the inter-stripe region between *eve* stripes 1 and 2. This transient inter-stripe expression lasts until 30–35 min into nc14; while inter-stripe expression between *eve* stripe 1 and *eve* stripe 2 disappears after ~20 min in wild-type embryos (compare ***Figure 2B and C***; compare ***Figure 2—figure supplement 1A and B***). These eveS1wt-eveS2Gt⁻ embryos did not produce the robust anterior expansion of *eve* stripe 2 described for the eveS2-MRE alone (***Small et al., 1992***). We attribute this muted anterior expansion in eveS1wt-eveS2Gt⁻ embryos (***Figure 2C***) to the regulatory sequences not present in the

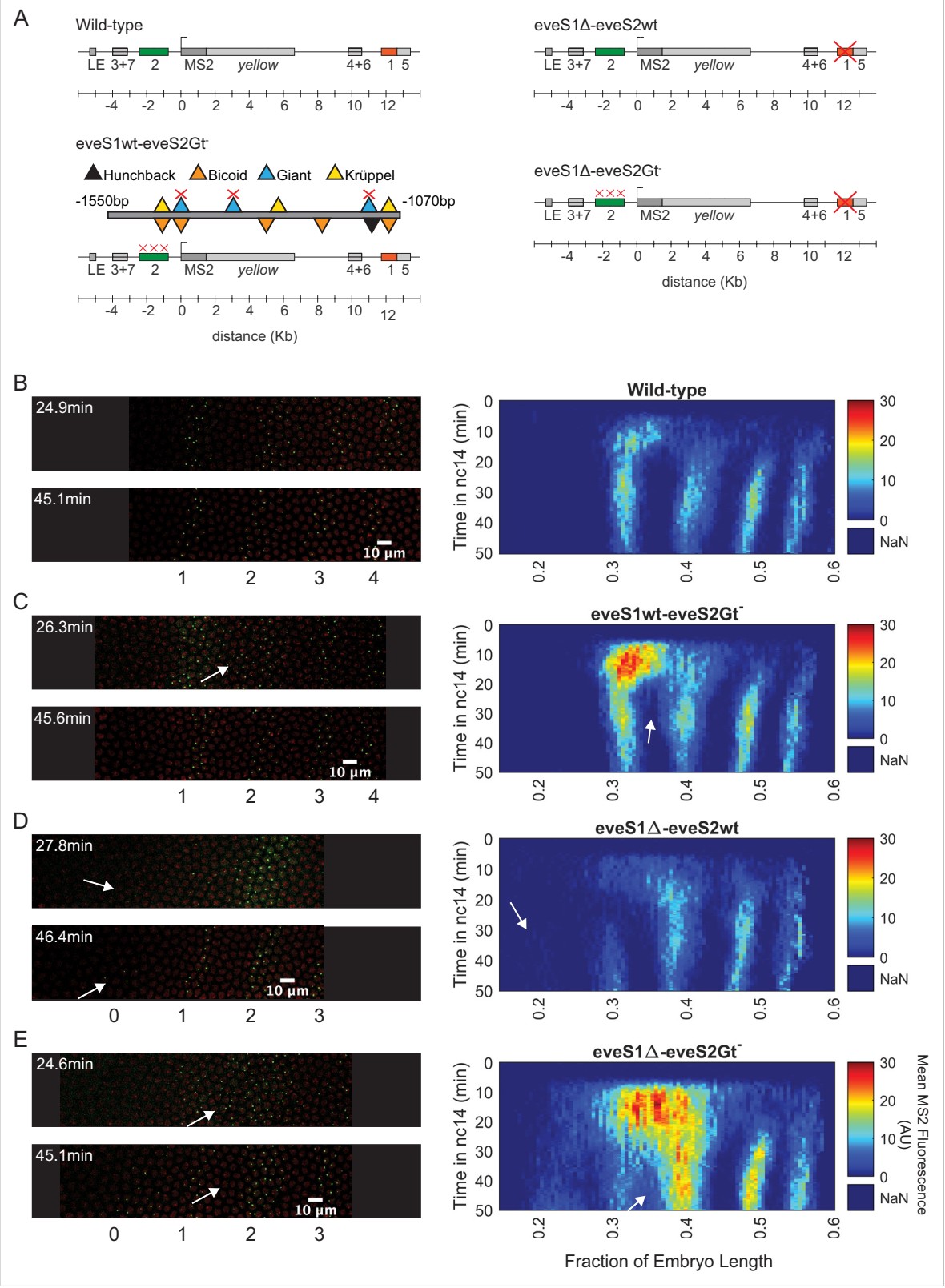

**Figure 2.** Transcriptional dynamics of eveMS2-BAC variants. (**A**) eveMS2 reporter construct variants used in this work. Boxes represent enhancers (e.g. *eve* stripe 2 enhancer is labeled as 2). LE is the *eve* late element. eveMS2-BAC is a reporter of wild-type *eve* expression. The eveS1wt-eveS2Gt⁻ carries a deletion of three Giant binding sites within the *eve* stripe 2 minimal regulatory element (eveS2-MRE; **Small et al., 1992**), as indicated by the three red crosses over the *eve* stripe 2 enhancer, and as shown in the detail of eveS2-MRE; where triangles represent transcription factor-binding sites. The

*Figure 2 continued on next page*

*Figure 2 continued*

eveS1Δ-eveS2wt carries a deletion of the *eve* stripe 1 enhancer. Finally, eveS1Δ-eveS2Gt⁻ combines the Giant binding site deletions from eveS1wt-eveS2Gt⁻ with the *eve* stripe 1 enhancer deletion of eveS1Δ-eveS2wt. (**B**) **Left**. Stills from a representative wild-type embryo at ~25 min and ~45 min into nuclear cycle 14 (nc14). Nuclei are labeled in red and transcription sites are labeled in green. **Right**. Kymograph of *eve* expression averaged over 5 eveMS2-BAC (wild-type) embryos. Time resolution along the y-axis is 20 seconds. The position of nuclei along the x-axis was calculated from various datasets, based on the inferred position of stripe centers, as described in the SI section: Generation of heatmaps in Figure 2 and *Figure 2—figure supplement 1* in Materials and methods. MS2 fluorescence in arbitrary units (AU) along the x-axis was averaged from nuclei located within bins of 0.5% embryo length. (**C**) **Left**. eveS1wt-eveS2Gt⁻ embryo at ~25 min and ~45 min into nc14. **Right**. Average *eve*-MS2 fluorescence from 6 eveS1wt-eveS2Gt⁻ embryos. At ~25 min, some transcriptionally active nuclei in the inter-stripe region between *eve* stripe 1 and *eve* stripe 2 can still be detected (white arrows), while, in wild-type embryos, *eve* stripe 1 and 2 are completely separated by ~20 min into nc14. (**D**) **Left**. eveS1Δ-eveS2wt embryo at ~25 min and ~45 min into nc14. **Right**. Average *eve*-MS2 fluorescence from 5 eveS1Δ-eveS2wt embryos. *eve* stripe 1 is almost absent at ~25 min, but appears later, probably driven by activity of the *eve* late element. A dim *eve* stripe 0 is apparent (white arrows). (**E**) **Left**. eveS1Δ-eveS2Gt⁻ embryo at ~25 min and ~45 min into nc14. **Right**. Average *eve*-MS2 fluorescence from 6 eveS1Δ-eveS2Gt⁻ embryos. At ~25 min, there is a strong ectopic expression in the inter-stripe region between *eve* stripe 1 and *eve* stripe 2 (white arrow). At ~45 min, this ectopic inter-stripe expression has dimmed (white arrows), while *eve* stripe 0 becomes apparent.

The online version of this article includes the following figure supplement(s) for figure 2:

**Figure supplement 1.** Spatiotemporal dynamics of *eve* expression across wild-type and mutant embryos in logarithmic scale.

original minimal *eve* stripe 2 reporter construct which might provide a buffering effect to the disruption of the three Giant binding sites (*López-Rivera et al., 2020*).

In an attempt to expand the anterior ectopic domain of eveS1wt-eveS2Gt⁻, we sought to free its expression domain from any potential interference from *eve* stripe 1 expression. To make this possible, we deleted endogenous expression corresponding to the *eve* stripe 1 enhancer. Specifically, we generated a mutant version of eveMS2-BAC with the *eve* stripe 1 enhancer deleted (eveS1Δ-eveS2wt; *Figure 2A and D*; *Figure 2—figure supplement 1C*). Unexpectedly, these embryos still exhibited a dim *eve* stripe 1 (~30% of embryo length) after ~30 min into nc14, perhaps due to the activity of the *eve* late element; and a dim additional anterior stripe that we refer to as *eve* stripe 0 (~20% embryo length) after ~25 min into nc14. In a previous study, *Small et al., 1992* identified a 'head patch' of gene expression when assaying the regulation of the minimal regulatory element of the *eve* stripe 2 enhancer. It is tempting to identify our *eve* stripe 0 with their observed head patch. (*Small et al., 1992*) speculated that this head patch was the result of sequences in the P-transposon system used for their genomic insertions, which are not present in our experimental design. Thus, the appearance of *eve* stripe 0 indicates a repressive role of *eve* stripe 1 enhancer beyond the anterior boundary of *eve* stripe 1 (*Figure 2D*), and it may imply that the minimal regulatory element of the *eve* stripe 2 enhancer can indeed drive expression in this head patch when *eve* stripe 1 enhancer is not present.

Finally, we coupled the three deletions of Gt-binding sites in the *eve* stripe 2 enhancer from eveS1wt-eveS2Gt⁻ with the complete deletion of the *eve* stripe 1 enhancer in eveS1Δ-eveS2wt to create eveS1Δ-eveS2Gt⁻ (*Figure 2A and E*; *Figure 2—figure supplement 1D*). Surprisingly, eveS1Δ-eveS2Gt⁻ embryos revealed large ectopic regions of *eve* expression more complex than the sum of patterns displayed by the independent mutants described above. Beyond a stronger and longer-lasting inter-stripe expression between *eve* stripe 1 and *eve* stripe 2 than observed in eveS1wt-eveS2Gt⁻, eveS1Δ-eveS2Gt⁻ embryos exhibited the following notable features: a stronger-than-wild-type *eve* stripe 2 (located at ~40% of embryo length); the presence of *eve* stripe 1 (~30% of embryo length) and *eve* stripe 0 (~20% of embryo length); and many *eve*-active nuclei in normally silent inter-stripe regions between *eve* stripe 2 and *eve* stripe 0 (*Figure 2E*). The fact that the knock-out of *eve* stripe 1 enhancer coupled with the disruption of Gt-binding sites in *eve* stripe 2 enhancer renders more ectopic expression on the anterior half of fruit fly embryos than the independent disruptions in eveS1Δ-eveS2wt and eveS1wt-eveS2Gt⁻ implies that the repressive activity of the *eve* stripe 1 enhancer synergizes with the repression exerted by Giant—and potentially with other unidentified transcription factors that bind in the vicinity of Gt-binding sites—on the *eve* stripe 2 enhancer. The hypothesis that Gt binding sites in *eve* stripe 2 enhancer may recognize other transcription factors was proposed by *Small et al., 1992*, who observed that the anterior expansion of *eve* stripe 2 that results from disrupting Gt-binding sites in *eve* stripe 2 enhancer is 'somewhat more severe' than the expansion observed in Gt⁻ embryos.

Taken together, our results suggest that the *eve* stripe 1 enhancer plays a repressing role in the anterior half of fruit fly embryos which synergizes with the Giant repressor and likely with other

transcriptional regulators bound to Gt binding sites or their vicinity in the *eve* stripe 2 enhancer. This argues in favor of cross-activity between the *eve* stripe 1 and 2 enhancers that impacts *eve* expression in the anterior half of the embryo. *eve* stripe 1 enhancer might be also playing a role in the regulation of *eve* stripe 2, as Giant-binding site deletions in the *eve* stripe 2 enhancer alone do not result in the stronger-than-wild-type *eve* stripe 2 observed in eveS1Δ-eveS2Gt⁻ embryos. In summary, coupling the disruption of Giant-binding sites in the *eve* stripe 2 enhancer with the deletion of the *eve* stripe 1 enhancer produces different mutant patterns than the sum of the individual mutants. Finally, regardless of the complex regulatory interactions uncovered by our enhancer mutants, our results indicate that the ectopic gene expression patterns driven by our eveS1Δ-eveS2Gt⁻ reporter provide an ideal scaffold for our investigations of the regulation of transcriptional bursting outside of endogenous embryo regions.

## Bursting strategies are uniform across endogenous and ectopic *eve*-active nuclei

We determined the position of nuclei displaying active *eve* transcription and labeled them as endogenous if they were positioned within the boundaries of wild-type *eve* stripes (*eve* stripe 1, *eve* stripe 2, *eve* stripe 3, *eve* stripe 4); or as ectopic if they were located in the inter-stripe region between *eve* stripe 1 and *eve* stripe 2 (*eve* stripe 1–2) or in *eve* stripe 0 (in the far anterior; *Figure 3A*) as described in Materials and methods. *eve* stripe 1 expression in embryos with disrupted *eve* stripe 1 enhancer was considered endogenous, as we believe that this expression results from activity of the late element. All active nuclei in wild-type embryos were labeled as endogenous. Overall, ectopic regions show lower levels of mean MS2 fluorescence than endogenous regions, as is evident by comparing *eve* the interstripe 1–2 and *eve* stripe 0 against *eve* stripe 1, *eve* stripe 2, and *eve* stripe 3 in eveS1Δ-eveS2Gt⁻ embryos (*Figure 2E*, *Right*). This is perhaps due to the unavailability of optimal concentrations of transcription factors; for example a lack of activators or an excess of repressors with respect to the concentrations found in endogenous regions (*Figure 1C and D*).

To uncover which bursting parameters are modulated to create each endogenous and ectopic stripes and interstripe regions, we need to extract the bursting parameters in each region. We computed bursting parameters for nuclei grouped by stripe and binned by transcriptional output (*Figure 3—figure supplement 1*) in our four fly lines, with the following rationale. In the bursting model, the average rate of transcription initiation is described by the formula $r \frac{kon}{kon+koff}$, where $\frac{kon}{kon+koff}$ indicates the fraction of time the promoter spends in the ON state (*Lammers et al., 2020*). As enhancers and their inputs (e.g. transcription factors, chromatin state) define bursting parameters ($k_{on}$, $k_{off}$, $r$), nuclei of similar average transcriptional output within the same stripe should be driven by similar inputs acting over the same enhancer. Thus, these nuclei should show similar values of the bursting parameters $k_{on}$, $k_{off}$ and $r$ that satisfy the equation above. On the other hand, our model predicts that nuclei with different *fluorescence* must differ in at least one of their bursting parameter values ($k_{on}$, $k_{off}$ and/or $r$).

The average MS2 fluorescence is a direct reporter of the average rate of transcriptional initiation. Single-cell MS2 fluorescence measurements reflect the transcriptional dynamics of individual promoters as they undergo transcriptional bursting (*Figure 3B*). However, the actual promoter states, or bursting parameters, underlying the transcriptional bursting remain 'hidden', as RNA Pol II molecules engage in elongation for several minutes (~140 s for the *MS2::yellow* transcriptional unit in our reporter system) (*Berrocal et al., 2020*). As a result, MS2 fluorescence is observable even after the promoter switches to the OFF state, convolving the promoter switching dynamics with those of transcriptional elongation. Thus, we can only compute promoter states by inferring them from MS2 fluorescence over time. To infer hidden promoter states, we used a compound-state Hidden Markov Model (cpHMM) developed by *Lammers et al., 2020*. By inferring the succession of activity states, cpHMM estimates rates of transitioning between the OFF and ON states ($k_{on}$ and $k_{off}$) and the rate at which ON promoters load active RNA Pol II molecules ($r$).

Consistent with our previous work (*Berrocal et al., 2020*), we find that endogenous stripes in eveMS2-BAC wild-type embryos modulate their transcriptional output (mean MS2 fluorescence in wild-type embryos ranges from 2 to 15 AU) by tuning the average $k_{on}$ (from 0.5 to 1.5 OFF to ON promoter transitions per minute) and $r$ (from an average fluorescence increase at a rate of 5 AU per minute to 10 AU per minute). The average $k_{off}$ value remains largely constant (0.5 ON to OFF promoter transitions per minute), with only a minor downregulation at high transcription outputs (*Figure 3C*).

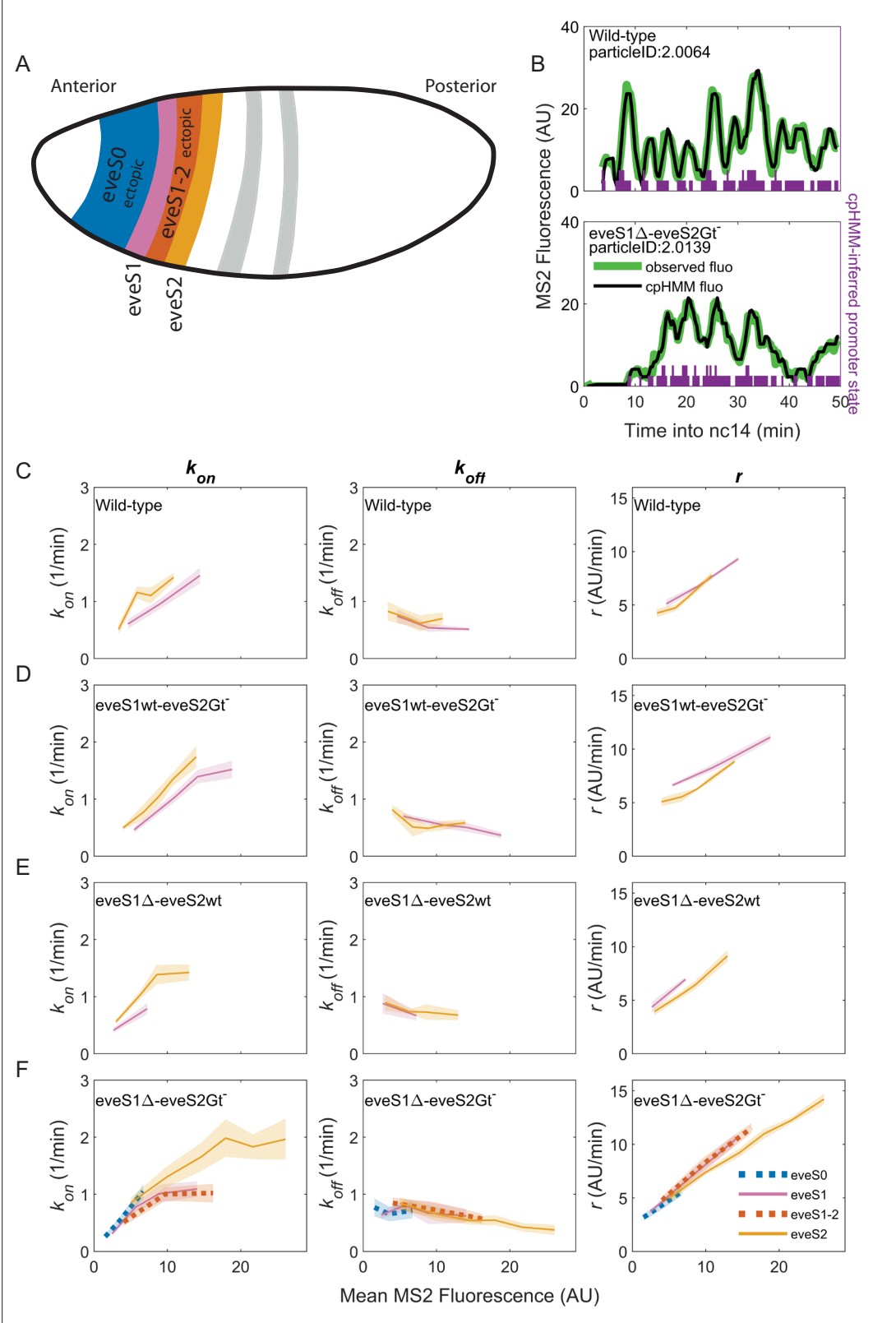

**Figure 3.** Bursting parameter control is almost identical in endogenous and ectopic gene expression regions. $k_{on}$ (*left panels*), $k_{off}$ (*middle panels*) and $r$ (*right panels*) trends across stripes, estimated from nuclei binned by their mean MS2 fluorescence. (**A**) Position and color code of endogenous and ectopic stripes in the fruit fly embryo. Only *eve* stripe 0, 1, 1–2, and 2 are shown for clarity. *Figure 3—figure supplement 2* includes *eve* stripe 3, and 4. (**B**) MS2 fluorescent traces (green) from embryos of different genotypes and cpHMM fit (black). Transcription in *Drosophila* embryos occurs after DNA

*Figure 3 continued on next page*

*Figure 3 continued*

replication. Since replicated sister chromatids remain paired, each *eve* locus contains two promoters, and every one of them can be ON or OFF. Purple bars show cpHMM-inferred promoter state corresponding to the two sister chromatids within a transcription spot (*Lammers et al., 2020*). Absence of bars represents both sister promoters OFF; shorter bars represent 1 sister promoter ON; longer bars represent 2 sister promoters ON. We aggregated the active state of 1 and 2 sister promoters into a single ON state, which leads to an effective two-state model of promoter activity (see SI section: Inference of Bursting Parameters in Materials and methods). Each point in the plots below was computed from ~40 fluorescent traces. (**C**) As previously observed in *eve*-MS2 wild-type embryos (*Berrocal et al., 2020*), nuclei in all stripes follow the same trends in bursting parameters. $k_{on}$, the average rate at which the promoter switches from OFF to ON increases with increasing transcriptional initiation as reported by MS2 fluorescence. $k_{off}$, the average rate at which a promoter switches from ON to OFF remains largely constant, and has a slight decrease in nuclei with the highest MS2 fluorescence values. $r$, the average rate at which active promoters increase their fluorescence, is higher in brighter nuclei. All stripes from (**D**) eveS1wt-eveS2Gt⁻ and (**E**) eveS1Δ-eveS2wt share the same bursting strategy. (**F**) The same trends occur in endogenous (eveS1 and eveS2; solid lines) and ectopic stripes (eveS0 and eveS1-2; dotted lines) of eveS1Δ-eveS2Gt⁻ embryos.

The online version of this article includes the following figure supplement(s) for figure 3:

**Figure supplement 1.** Pipeline for the quantification of *eve* bursting parameters ($k_{on}$, $k_{off}$, $r$) in nuclei grouped by stripe and binned by mean MS2 fluorescence (*Figure 3*).

**Figure supplement 2.** Bursting parameter inference for all stripes recorded in our data.

**Figure supplement 3.** Comparison of bursting parameters between endogenous and ectopic *eve* expression regions.

Thus, we confirm that *eve*-active nuclei in all wild-type stripes achieve higher levels of transcription by upregulating average bursting frequency ($k_{on}$) and amplitude ($r$), while average burst duration ($k_{off}^{-1}$) remains largely the same.

eveS1wt-eveS2Gt⁻ (*Figure 3D*) and eveS1Δ-eveS2wt (*Figure 3E*) embryos did not yield enough ectopic nuclei for cpHMM inference. However, their endogenous stripes followed the same bursting strategy observed in wild-type embryos, regardless of whether stripes were activated by wild-type or mutant enhancers (see SI Section: Complementary analysis of bursting parameters in Materials and methods). We inferred bursting parameters across regions of endogenous and ectopic nuclei in eveS1Δ-eveS2Gt⁻ embryos (*eve* stripe 1–2 and *eve* stripe 0), as they yielded sufficient ectopic *eve*-active nuclei to support cpHMM inference. As noted previously, these embryos feature an *eve* stripe 2 with nuclei of higher transcriptional output than wild-type embryos (compare *Figure 2B and E*), and a large region of ectopic expression towards the embryo anterior. Despite these differences in transcriptional output, bursting parameters in endogenous and ectopic *eve*-active nuclei in eveS1Δ-eveS2Gt⁻ embryos follow the same trends as wild-type embryos (*Figure 3—figure supplement 2*). In all regions–both endogenous and ectopic–enhancers increase transcription by increasing in $k_{on}$ and $r$, while $k_{off}$ remains largely constant (*Figure 3F*).

We performed an orthogonal cpHMM inference of bursting parameters by grouping nuclei in only two categories (endogenous and ectopic) (*Figure 3—figure supplement 3*), instead of grouping them according to their stripe, and we observed that this approach renders the same results (see SI Section: Complementary analysis of bursting parameters in Materials and Methods).

Taken together, our results show that all *eve* enhancers modulate their transcriptional output by increasing burst frequency ($k_{on}$) and amplitude ($r$). $k_{off}$, which shapes burst duration, remains largely constant, and shows a subtle drop as the mean MS2 fluorescence of nuclei increases. A wide range of transcriptional outputs result from these parameters. *eve* strategies of bursting control are robust to mutations on *eve* enhancers, and remain consistent in the presence of a myriad of inputs, including ectopic inputs different from those that shape the transcription of the seven canonical *eve* stripes.

## Discussion

Over the last few years, the ability to infer bursting parameters from fixed (*Little et al., 2013*; *Xu et al., 2015*) and live-imaging (*Lammers et al., 2020*) data in embryos has revealed several common-alities and differences in the strategies employed by different enhancers to modulate bursting param-eters and create patterns of gene expression (*Berrocal et al., 2020*; *Zoller et al., 2018*). For example, despite the different inputs that regulate the activity of *eve* enhancers, all of them modulate the expression of the seven canonical *eve* stripes by upregulating burst frequency ($k_{on}$) and amplitude ($r$), while burst duration ($k_{off}^{-1}$) remains largely constant and shows only a minor increase in nuclei of high transcriptional output (*Berrocal et al., 2020*). Since the seven *eve* stripes are largely controlled

by independent enhancers that respond to unique combinations of transcription factors, it was still unclear whether *eve* enhancers employ the same bursting strategy in ectopic regions, in the presence of *trans*-regulatory environments different from those that exist in their wild-type regions of expression.

Different bursting strategies between endogenous and ectopic regions of *eve* expression would suggest a selective pressure on *eve* enhancers that favors the observed bursting strategies at their canonical expression domains. On the other hand, unified bursting strategies in endogenous and ectopic regions point towards a common molecular mechanism, constrained by the biochemistry of enhancer-promoter interaction, which shapes the observed bursting parameters independent of changing *trans*-regulatory environments.

In this work, we compared bursting parameters ($k_{on}$, $k_{off}$, $r$) between endogenous and ectopic regions of *eve* expression to test between those two hypotheses. Specifically, we performed live imaging of *eve*-enhancer activity and bursting parameter inference in *D. melanogaster* embryos expressing wild-type and mutant versions of our BAC-based eveMS2 reporter system. Our observations provide evidence in favor of the second hypothesis, as we observe a unified strategy of bursting control wherever *eve* enhancers are active, regardless of the ectopic or endogenous inputs that regulate their activity. However, we acknowledge that our work cannot conclusively rule out the possibility that the observed strategies of bursting control may have been selected by evolution as the most optimal for the expression of the seven endogenous *eve* stripes. In this scenario, bursting control strategies would be conserved in ectopic expression regions as an evolutionarily neutral 'passenger phenotype'. Regardless, the novelty of our current work lies in the insights derived from the comparative analysis of bursting control strategies between ectopic and endogenous *eve* expression regions, an aspect not addressed in *Berrocal et al., 2020*. In summary, despite changing *trans*-regulatory environments and mutations in enhancer sequence, *eve* enhancers act through a single promoter and upregulate transcriptional bursting in endogenous and ectopic expression regions. It is important to note that the modulation of burst frequency and amplitude is not the only possible bursting control strategy, and we emphasize that the unified strategies of *eve* bursting control described in this study do not necessarily apply to other genes. Indeed, (*Zoller et al., 2018*) observed that *Drosophila* gap genes, controlled by independent promoters and enhancers, modulate bursting through another common strategy; an increase in frequency and duration, while burst amplitude remains unchanged. A subsequent study by P.-T. *Chen et al., 2023* found further evidence of a tight relationship between burst frequency and duration among gap genes. Consistent with our findings on *eve* bursting control, the authors observed that bursting control strategies for gap genes persist despite genetic perturbations. Furthermore, in a recent study, *Syed et al., 2023* utilized a Hidden Markov Model to analyze live imaging data of transcription driven by *snail* enhancers. The study concludes that disrupting Dorsal binding sites on the *snail* minimal distal enhancer leads to a reduction in both the amplitude and duration of transcription bursting in fruit fly embryos. This work underscores the significance of enhancer-transcription factor interactions in shaping the bursting strategies of *snail* gene. These findings hint at an opportunity to classify enhancers and promoters in families whose members employ the same strategy of bursting control and rely on a common molecular mechanism to regulate their target genes.

In the light of our results, two molecular mechanisms coupled to enhancer activity could be behind the unified bursting strategies of *eve* enhancers. First, the observed common modulation of bursting parameters might result from general constraints imposed by the transcriptional machinery at enhancers or promoters. Previous work showed that topological dynamics of bacterial chromosomes brought by transcriptional activity shape bursting in bacteria (*Chong et al., 2014*); while histone acetylation of the circadian promoter *Bmal1* modulates burst frequency in mammalian cells (*Nicolas et al., 2018*). Furthermore, *Gorski et al., 2008* observed that the dynamics of RNA Pol I–subunit assembly affect transcriptional output. The dynamic nature of transcription factor 'hubs' (*Mir et al., 2017*; *Tsai et al., 2017*) in transcriptionally active enhancers of *D. melanogaster* embryos (*Mir et al., 2018*) may impact transcriptional bursting as well. The importance of modulating the concentration and availability of key transcription factors is emphasized by *Hoppe et al., 2020*. Their findings show that the naturally established concentration gradient of Bone Morphogenetic Protein (BMP) defines the bursting frequency of BMP target genes in fruit fly embryos. Another example that underscores the significance of transcription factor availability in shaping bursting strategies was

illustrated by *Zhao et al., 2024*. Using optogenetic LEXY-mediated modulation of nuclear protein export (*Niopek et al., 2016*) in fruit fly embryos, this study found that the transcription factor Knirps represses the activity of the *eve* stripe 4+6 enhancer by gradually decreasing burst frequency until the locus sets into a fully reversible quiescent state. Systematic modulation of nuclear concentration through optogenetic LEXY for critical transcription factors such as Bicoid, Hunchback, Giant, Kruppel, and Zelda, will aid in fully elucidating the impact of transcription factor dynamics on *eve* bursting control strategies.

The second possibility is that the *eve* promoter, which is shared by all *eve* enhancers and distant regulatory elements, constrains the regulatory strategy of *even-skipped*. Recent studies using MS2 live imaging have described a fundamental role of core promoter elements, such as the TATA box, the initiator element, and the downstream core promoter element in shaping transcriptional bursting in genes of *D. melanogaster* embryos (*Pimmett et al., 2021*; *Yokoshi et al., 2022*). Furthermore, a survey of 17 genes in the actin family of the amoeba *Dictyostelium discoideum* (*Tunnacliffe et al., 2018*), featuring identical coding sequences but distinct promoters, revealed different bursting behaviors for each gene. These observations hint at a critical role of promoters in shaping bursting strategies. Further experiments, exploring the bursting strategies that result from swapping promoters in constructs carrying the *eve* enhancers could elucidate whether the *eve* promoter is responsible for establishing the *eve* regulatory strategy.

Both possibilities suggest that a molecular mechanism coupled to *eve* transcription restricts the landscape of bursting strategies available to *eve* enhancers. Our results indicate that *eve* bursting strategies are a fundamental property of enhancers and promoters—and not the result of changing *trans*-regulatory environments—and show that *eve* enhancers merely act as knobs, robust to mutations, that tune transcriptional output levels by modulating bursting through a largely fixed $k_{off}$ and shifting $r$ and $k_{on}$.

An ectopic pattern of particular interest is the novel *eve* stripe 0 brought by the deletion of the *eve* stripe 1 enhancer. This new stripe shows that mutations on existing *eve* enhancers can generate novel gene expression patterns through the same bursting strategies employed by the other *eve* stripes. Since expression patterns in embryonic development shape the formation and identity of animal body plans (*Akam, 1983*; *Davidson, 2010*; *Lewis, 1978*), the appearance of new expression patterns may constitute a critical driver of evolution (*Rebeiz et al., 2011*).

## Materials and methods
### DNA constructs and fly lines

We generated four reporter constructs based on a previously established Bacterial Artificial Chromosome (BAC) carrying the ~20 Kb DNA sequence around *eve* (*Venken et al., 2006*; *Venken et al., 2009*), and whose *eve* coding sequence has been replaced by an MS2::*yellow* transcriptional unit (*Berrocal et al., 2020*). We used wild-type eveMS2-BAC from *Berrocal et al., 2020*. The other three BAC constructs were derived from wild-type eveMS2-BAC. These constructs carried mutant versions of *eve* stripe 1 and *eve* stripe 2 enhancers. Vector Builder (https://en.vectorbuilder.com/) generated the mutant versions through ccdB-amp cassette mediated recombineering. These mutant BACs are available on Vector Builder's website. SnapGene (.dna) files with eveMS2 BAC sequences are in the repository https://github.com/aberrocal/BurstingStrategies-eve, folder BurstingStrategies-eve/_DataSubmission/BACSequences/.

### eveS1wt-eveS2Gt⁻

BAC construct (Vector Builder-Service Proposal: P180328-1009dgs) contains a wild-type *eve* stripe 1 and a mutant version of *eve* stripe 2 enhancer with three Giant-binding sites deleted, as shown in Table 1 of *Small et al., 1992*. We chose to disrupt the three Gt-binding sites within the *eve* stripe 2 enhancer (*Figure 2B*) that had previously been tied to ectopic anterior expansion of *eve* stripe 2 expression when deleted in the context of the Minimal Regulatory Element of the eveS2 enhancer (eveS2-MRE; *Small et al., 1992*). eveS2-MRE is a 480 bp regulatory sequence within the *eve* stripe 2 enhancer (~2 kb total length) sufficient to drive the expression of *eve* stripe 2.

### eveS1Δ-eveS2Gt⁻

BAC construct (Vector Builder-Service Proposal: P180614-1002pzr) has the *eve* stripe 1 enhancer, as defined by ChIP-seq data of the enhancer-associated protein Zelda (*Harrison et al., 2011*), replaced by a ccdB-amp cassette and *eve* stripe 2 enhancer replaced by a mutant version with three Giant binding sites deleted as described above.

### eveS1Δ-eveS2wt

BAC construct (Vector Builder-Service Proposal: P190605-1001zkt) has *eve* stripe 1 enhancer replaced with a ccdB-amp cassette and a wild-type *eve* stripe 2. To sum up, we used the fly line carrying wild-type eveMS2-BAC from *Berrocal et al., 2020* and we generated three new fly lines carrying genome integrations of the aforementioned constructs. The mutant versions of eveMS2-BAC used in this work were inserted in the genome via φC31 integrase-mediated recombination. Mutant constructs were either sent to BestGene Inc (eveS1wt-eveS2Gt⁻, eveS1Δ-eveS2wt) for germline injection or injected in our laboratory (eveS1Δ-eveS2Gt⁻). All constructs integrated into a φC31 AttP insertion site in chromosome 3 L (Bloomington stock #24871; landing site VK00033; cytological location 65B2).

**Table 1.** Datasets and stripes.
We recorded 5 wild-type eveMS2-BAC (eveS1wt-eveS2wt) datasets, 6 eveS1wt-eveS2Gt⁻ (eveS1wt_eveS2Gt), 5 eveS1Δ-eveS2wt (eveS1Null_eveS2wt), and 6 eveS1Δ-eveS2Gt⁻ (eveS1Null_eveS2Gt) for a total of 22 datasets. Movies in every dataset capture between 3 and 6 stripes. Table 1 shows stripes captured in each dataset. Stripes in parentheses had few active nuclei (eveS0) or were not captured in their entirety (eveS4) and (eveS5). Asterisks indicate datasets used for stills in *Figure 2*.

| Wild-type datasets | Stripes Recorded |
| --- | --- |
| eveS1wt_eveS2wt_1 | eveS1, eveS2, eveS3, eveS4 |
| eveS1wt_eveS2wt_2 | eveS1, eveS2, eveS3, (eveS4) |
| eveS1wt_eveS2wt_3* | eveS1, eveS2, eveS3, eveS4 |
| eveS1wt_eveS2wt_4 | eveS1, eveS2, eveS3, eveS4 |
| eveS1wt_eveS2wt_5 | eveS1, eveS2, eveS3, eveS4, (eveS5) |
| **eveS1wt-eveS2Gt⁻ datasets** | **Stripes Recorded** |
| eveS1wt_eveS2Gt_1 | eveS1, eveS1-2, eveS2, eveS3 |
| eveS1wt_eveS2Gt_2 | eveS1, eveS1-2, eveS2, eveS3, (eveS4) |
| eveS1wt_eveS2Gt_3 | eveS1, eveS1-2, eveS2, eveS3, eveS4 |
| eveS1wt_eveS2Gt_4 | eveS1, eveS1-2, eveS2, eveS3 |
| eveS1wt_eveS2Gt_5* | eveS1, eveS1-2, eveS2, eveS3, eveS4 |
| eveS1wt_eveS2Gt_6 | eveS1, eveS1-2, eveS2, eveS3, eveS4 |
| **eveS1Δ-eveS2wt datasets** | **Stripes Recorded** |
| eveS1Null_eveS2wt_1 | (eveS0), eveS1, eveS2, eveS3, eveS4 |
| eveS1Null_eveS2wt_2* | eveS0, eveS1, eveS2, eveS3 |
| eveS1Null_eveS2wt_3 | eveS0, eveS1, eveS2, eveS3, (eveS4) |
| eveS1Null_eveS2wt_4 | (eveS0), eveS1, eveS2, eveS3, (eveS4) |
| eveS1Null_eveS2wt_5 | eveS0, eveS1, eveS2, eveS3 |
| **eveS1Δ-eveS2Gt⁻ datasets** | **Stripes Recorded** |
| eveS1Null_eveS2Gt_1 | eveS0, eveS1, eveS1-2, eveS2, eveS3, eveS4 |
| eveS1Null_eveS2Gt_2 | eveS0, eveS1, eveS1-2, eveS2, eveS3, eveS4 |
| eveS1Null_eveS2Gt_3 | eveS0, eveS1, eveS1-2, eveS2, eveS3, eveS4 |
| eveS1Null_eveS2Gt_4 | eveS0, eveS1, eveS1-2, eveS2, eveS3, eveS4 |
| eveS1Null_eveS2Gt_5* | eveS0, eveS1, eveS1-2, eveS2, eveS3 |
| eveS1Null_eveS2Gt_6 | eveS0, eveS1, eveS1-2, eveS2, eveS3 |

## Imaging

We crossed male flies from lines carrying eveMS2-BAC constructs (w-; +; MS2::yellow) and female flies carrying His::RFP and MCP::GFP fusion proteins (yw; His::RFP; MCP::GFP; *Garcia et al., 2013*). His::RFP allows for visualization of nuclei, MCP::GFP binds MS2 nascent transcripts to form fluorescent puncta at sites of nascent MS2 transcription. We set embryo-collection cages with ~30 male and ~100 female fruit flies, and collected offspring embryos after 1 hr 30 min. All movies in the same dataset were recorded within ~1 week. We mounted embryos on a slide for confocal imaging, as described in *Berrocal et al., 2020*; *Bothma et al., 2014*. Aging embryos for 1 hr 30 min allows us to capture the entire interval between the 14th synchronous cell cleavage and the beginning of gastrulation. We recorded a total of 22 live embryos as shown in *Table 1*. All imaging was done in a Zeiss-800 scanning-laser confocal microscope. Movies of embryonic development were captured under a 63 x oil objective, in windows of 202.8 µm x 50.7 µm, at pixel size of 0.2 µm, zoom 0.5 x. Movies were recorded in two channels, EGFP for MS2 signal, and TagRFP for His::RFP signal. Imaging parameters were 16 bits per pixel, scan mode frame, bidirectional scanning, scan speed 7, pixel dwelling 1.03 µs, laser scanning averaging 2, averaging method mean, averaging mode line, laser power EGFP 30 µW and TagRFP 7.5 µW, master gain in EGFP channel 550 V and in TagRFP channel 650 V, digital offset in both channels 0, digital gain in both channels 1, pinhole size 44 µm (1 Airy unit - 0.7 µm/section) at 63 x objective, laser filters EGFP:SP545 and TagRFP:LBF640. Data points consist of Z-stacks of 21 slices separated by intervals of 0.5 µm, to span a range of 10 µm across the Z axis. Z-stack mode full stack. Whole Z-stacks were recorded every 16.8 s (wild-type, eveS1wt-eveS2Gt⁻, eveS1Δ-eveS2Gt⁻) and 19.5 s (eveS1Δ-eveS2wt). The difference in time resolution between datasets does not impact our analysis, as the cpHMM analyzes interpolated data points at 20 s intervals. These parameters are based on the imaging protocol and settings in *Berrocal et al., 2020*. We stopped live imaging of individual embryos after 50 min into nuclear cycle 14, before the cell rearrangements of gastrulation, and took mid-sagittal and surface images of the whole embryo to localize our 202.8 µm x 50.7 µm window along the embryonic anterior-posterior axis. Raw data from confocal microscope imaging is publicly available in Zenodo (https://zenodo.org/, https://doi.org/10.5281/zenodo.7204096; see SI section: Data and Code) (*Table 1*).

## Segmentation and quantification of movies

We tracked MS2 foci from movies and segmented them using the MATLAB based analysis pipeline developed by *Berrocal et al., 2020*; *Garcia et al., 2013*; *Lammers et al., 2020*. Specifically, for segmentation of MS2/MCP::GFP foci across stacks on the Z-axis, we combined the MATLAB pipeline mentioned above with Fiji-Weka Segmentation 3D software, as described in *Berrocal et al., 2020*. The MATLAB/Fiji-Weka pipeline extracts the position of nuclei and the fluorescence intensity and position of individual MS2 foci over time. The final result of the MATLAB based analysis pipeline are CompiledParticles.mat files that contain the position of nuclei, as well as their MS2 fluorescence intensity over time (see Data and Code).

## Assignment of *eve*-active nuclei to stripes

We manually segmented nuclei from eveS1Δ-eveS2Gt⁻ and eveS1wt-eveS2Gt⁻ fly lines, as their stripes were not always clearly discernible. For these embryos, we assigned nuclei to individual stripes based on the position of stripes at 45 min into nc14, when they became separated from the background. The boundary between *eve* stripe 1–2 and *eve* stripe 2 in eveS1Δ-eveS2Gt⁻ embryos was set at 36% of embryo length, according to the kymograph of MS2 fluorescence over time. On the other hand, eveS1Δ-eveS2wt and wild-type embryos showed defined stripes after 25 min into nc14. Thus, we used a MATLAB k-means clustering algorithm to dynamically assign *eve*-active nuclei to individual stripes, tracking nuclei by the accumulation of MS2 fluorescent output in windows of five-minutes. Nuclei active between 0 and 25 min into nc14 were assigned to stripes based on their position at 25 min into nc14. We generated movies of segmented MS2 spots assigned to individual stripes in windows of ~5 minutes. MATLAB scripts for manual and k-means-automated segmentation of stripes, as well as scripts to generate movies of segmented stripes are available in github (see Data and Code).

## Generation of heatmaps in Figure 2 and Figure 2-Figure Supplement 1

We used traces of MS2 fluorescence intensity over time, which reflect transcriptional activity, to generate heatmap/kymographs of MS2 transcription datasets. We generated heatmaps (*Figure 2*, *Figure 2—figure supplement 1*) by collapsing data points from all embryos of the same genotype into a single kymograph plot. We started by adjusting the position of nuclei in each embryo relative to nuclei in other embryos of the same genotype. As we had assigned MS2 active nuclei to individual stripes, we measured the distance along the anterior-posterior axis from each MS2 focus to the center of its corresponding stripe. We inferred the position of pseudo-stripes formed by the combined data from all embryos of the same genotype. We calculated the position of pseudo-stripes along the anterior-posterior embryo axis by averaging the position of the center of stripes along the anterior-posterior axis in individual embryos of the same genotype. Finally, we assigned a position to all nuclei of the same genotype relative to pseudo-stripes by positioning them at the same distance from the center of pseudo-stripes as they were from the center of the stripe where they originated. We followed the same procedure to locate the position of inactive nuclei.

## Labeling *eve* patterns as endogenous or ectopic

To compare the bursting parameters between endogenous and ectopic regions of *eve* activity, we segmented MS2-active nuclei and assigned them to individual regions that were deemed to be either endogenous or ectopic. We labeled regions as endogenous if their position overlapped within the boundaries of wild-type *eve* stripes (*eve* stripe 1, *eve* stripe 2, *eve* stripe 3, *eve* stripe 4); or as ectopic if their position overlapped with the inter-stripe region between *eve* stripe 1 and *eve* stripe 2 (*eve* stripe 1–2) or with the novel *eve* stripe 0 (~20% embryo length). All stripes in wild-type embryos were labeled as endogenous.

## Selection of a three-state model of promoter activity and a compound Hidden Markov Model for inference of promoter states from MS2 fluorescent signal

We selected a three-state model of promoter activity (OFF, $ON_1$, $ON_2$) based on the following argument. Transcription in pre-gastrulating *Drosophila* embryos occurs after DNA replication, and sister chromatids remain paired. However, most of the time, paired MS2-tagged sister loci cannot be resolved independently using diffraction-limited microscopy (*Lammers et al., 2020*). Therefore, each fluorescent spot in our data results from the combined activity of two promoters, each of which, in the simplest possible model of transcriptional bursting, may be ON or OFF (*Lammers et al., 2020*). To account for this, the cpHMM infers three states from the observed MS2 data: OFF (both sister promoters inactive), $ON_1$ (one sister promoter active), and $ON_2$ (two sister promoters active). For ease of presentation, we aggregated $ON_1$ and $ON_2$ states into a single effective ON state, as we did in our previous work (*Berrocal et al., 2020*). This leads to an effective two-state model with one OFF and one ON state and three burst parameters: $k_{off}^{-1}$ (the burst duration), $k_{on}$ (the burst frequency), and $r$ (the burst amplitude). $k_{on}$ is defined as the sum of the transition rates from OFF to any of the two active states described above: OFF → $ON_1$ and OFF → $ON_2$. $k_{off}$ is defined as the rate at which the system returns to the OFF state upon leaving it, which is described by the formula $k_{off}^{-1} = (\frac{1}{p_{off}} - 1) k_{on}^{-1}$, where $p_{off}$ is the fraction of time the system spends in the OFF state. $k_{off}$ is the inverse of mean burst duration. $r$ is defined by the average of the rates of transcription initiation in the two ON states ($r_1$ and $r_2$) weighted by the fraction of the time that the system spends on each state ($p_1$ and $p_2$) as described by the formula $r = \frac{p_1\, r_1 + p_2\, r_2}{p_1 + p_2}$ (*Lammers et al., 2020*). The outputs of the three state model of promoter activity ($k_{on}$, $k_{off}$, and $r$) were used for downstream analyses.

The three-state model of promoter activity is the simplest model compatible with our current understanding of transcription at the *eve* locus in early fruit fly embryos. However, we do not dismiss the possibility that more complex processes, not captured by our model, define *eve* transcription. Promoters, for instance, may exhibit more than two states of activity, beyond a simple ON and OFF mechanism. Nevertheless, as pointed out by *Lammers et al., 2020* - SI Section: G. the cross-validation of cpHMM inference sensitivities between different model schemes (two, three, or multiple state Hidden Markov Models) do not yield consistent results regarding on which one is more accurate; and for the time being, there is no alternative to a HMM for inference of promoter states from MS2/PP7 fluorescence signals obtained using laser-scanning confocal microscopy (*Lammers et al., 2020*;

**Table 2.** Binning by stripe.

We pooled together nuclei from all embryos per dataset, sorted them by the stripe where they were located and distributed them in bins of varying fluorescence. Each bin contains ~40 nuclei (~2,500 time points). E.g., all nuclei in *eve* stripe 1 (eveS1) from the five *eve* wild-type embryos in our dataset were assigned to 3 bins according to their mean MS2 fluorescence, as each bin must contain ~40 nuclei, or ~2,500 data points, for input into the cpHMM.

| Wild-type - Stripes | Number of bins |
|---|---|
| eveS1 | 3 |
| eveS2 | 4 |
| eveS3 | 3 |
| eveS4 | 3 |
| eveS5 | 0 |
| **eveS1wt-eveS2Gt⁻ - Stripes** | **Number of bins** |
| eveS1 | 4 |
| eveS1-2 | 0 |
| eveS2 | 5 |
| eveS3 | 4 |
| eveS4 | 2 |
| **eveS1Δ-eveS2wt - Stripes** | **Number of bins** |
| eveS0 | 0 |
| eveS1 | 2 |
| eveS2 | 4 |
| eveS3 | 3 |
| eveS4 | 1 |
| **eveS1Δ-eveS2Gt⁻ - Stripes** | **Number of bins** |
| eveS0 | 3 |
| eveS1 | 4 |
| eveS1-2 | 3 |
| eveS2 | 6 |
| eveS3 | 5 |
| eveS4 | 3 |

*Syed et al., 2023*; although other approaches exist using state-of-the-art microscopy and deconvolution algorithms to improve signal-to-noise ratio). Furthermore, orthogonal approaches to quantify transcription that rely on static methods, such as smFISH, have a limited ability to capture temporal dynamics. Due to these considerations, we selected a HMM based on an effective two-state model (derived from a three-state model) of promoter activity to describe our live MS2 imaging data.

## Inference of bursting parameters

We used a cpHMM approach (*Lammers et al., 2020*) to extract average bursting parameters ($k_{on}$, $k_{off}$, $r$) from different sets of MS2-active nuclei. We input MS2 fluorescent traces over time from these sets into the cpHMM. Specifically, we combined nuclei from same-genotype embryos, sorted them by stripe and distributed them across bins of varying fluorescence. To ensure reliable inference, we enforced each bin to contain ~40 nuclei, equivalent to ~2500 time points at a 20 s resolution (*Lammers et al., 2020*). The number of bins was determined by the amount of data available (*Table 2*).

Wild-type embryos yielded sufficient nuclei to support the cpHMM inference of bursting parameters for various endogenous stripes (*eve* stripe 1, 2, 3, 4). eveS1wt-eveS2Gt⁻ and eveS1Δ-eveS2wt did not yield enough ectopically active nuclei for cpHMM analysis (*eve* stripe 1–2 in eveS1wt-eveS2Gt⁻;

**Table 3.** Binning by endogenous/ectopic.

We pooled together nuclei from all embryos per dataset, sorted them by endogenous or ectopic, according to whether the stripe where they were located was deemed endogenous or ectopic, and distributed them in bins of varying fluorescence. Each bin contains ~40 nuclei (~2500 time points). E.g. All endogenous nuclei in the 5 *eve* wild-type embryos were distributed among 11 bins of increasing MS2 fluorescence. Some datasets have their ectopic bin empty, as they had less than ~40 active nuclei in their ectopic regions.

| Wild-type | Number of Bins |
|---|---|
| Ectopic | 0 |
| Endogenous | 11 |
| eveS1wt-eveS2Gt⁻ | Number of Bins |
| Ectopic | 0 |
| Endogenous | 13 |
| eveS1Δ-eveS2wt | Number of Bins |
| Ectopic | 0 |
| Endogenous | 7 |
| eveS1Δ-eveS2Gt⁻ | Number of Bins |
| Ectopic | 6 |
| Endogenous | 11 |

*eve* stripe 0 in eveS1Δ-eveS2wt). These fly lines did exhibit endogenous *eve* stripes with enough active-nuclei for further analysis on the cpHMM (*eve* stripe 1, 2, 3, and 4 in eveS1wt-eveS2Gt⁻; *eve* stripe 1, 2, and 3 in eveS1Δ-eveS2wt). eveS1Δ-eveS2Gt⁻ embryos did yield sufficient *eve*-active nuclei (297 nuclei) to support cpHMM inference of the bursting parameters of ectopic *eve* stripe 1–2 and *eve* stripe 0. It also resulted in enough active nuclei for the cpHMM inference of bursting parameters of endogenous stripes (*eve* stripe 1, 2, 3, and 4).

The output of the effective two-state cpHMM described above are the bursting parameters ($k_{on}$, $k_{off}$, $r$) for each set of nuclei input into the model. Thus, *Figure 3*, *Figure 3—figure supplement 2* are plots of mean $k_{on}$, $k_{off}$, $r$, and their standard deviations $\sigma_{kon}$, $\sigma_{koff}$, $\sigma_r$, computed from sets of nuclei binned by stripe. For *Figure 3—figure supplement 3*, we followed a similar approach, but grouping active nuclei by their endogenous or ectopic location. Nuclei grouped in endogenous and ectopic categories were distributed across 6–13 bins of increasing fluorescence (*Table 3*). Their mean $k_{on}$, $k_{off}$, $r$, and standard deviations, $\sigma_{kon}$, $\sigma_{koff}$, $\sigma_r$ were plotted in *Figure 3—figure supplement 3*.

## Data and code

Raw data, Movies, and CompiledParticles files are stored in the Zenodo dataset 'Unified bursting strategies in ectopic and endogenous even-skipped expression patterns - Supplemental Data' (https://doi.org/10.5281/zenodo.7204096; *Berrocal et al., 2023*). Specific paths in this dataset are listed below. Raw confocal-imaging data from embryos of each of the genotypes used in this work are located in *[Genotype]_rawData/[Date]/[Dataset]* as czi files (Zeiss file format) of sequential Z-stacks recorded over two channels, and whole embryo stills, as described above. Maximum Z-projection movies of all recorded embryos are in *Movies/[Genotype]/Composite*. Movies of MS2-foci assigned to stripes are in *Movies/[Genotype]/Segmentation*. The outcome of *Garcia et al., 2013* MATLAB pipeline to analyze MS2 data from each embryo are mat files named CompiledParticles, they are stored in the folder *CompiledParticles/[Genotype]*.

MATLAB scripts and data for this analysis are stored in the github repository https://github.com/aberrocal/BurstingStrategies-eve. The code for the segmentation of our live imaging data of *eve* transcription in embryonic development is in BurstingStrategies-eve/_DataSubmission/DataSheetsAndCode/StripeSegmentation/. We generated csv files containing the position of active and inactive nuclei over time for each of four genotypes (see BurstingStrategies-eve/_DataSubmission/DataSheetsAndCode/Heatmaps/singleTraceFits_Heatmaps/). In these files, active nuclei have fluorescence values associated with each time point. These datasets also contain the promoter state of active nuclei

at each time point. We considered three promoter states: 1=OFF, 2=one sister promoter ON (ON$_1$), and 3=two sister promoters ON (ON$_2$); see SI section: Inference of bursting parameters in Materials and methods. The heatmaps in this work (*Figure 2*, *Figure 2—figure supplement 1*) were generated with MATLAB scripts and datasets in BurstingStrategies-eve/_DataSubmission/DataSheetsAndCode/ Heatmaps/. We generated mat files (*compiledResults_[Stripe/ectopicFlag].mat*) that contain mean values of $k_{on}$ (frequency), $k_{off}^{-1}$ (duration), $r$ (amplitude), their standard deviations, and mean fluorescence bin values. *compiledResults_Stripe.mat* files and scripts to generate *Figure 3*, *Figure 3—figure supplement 2* are sorted by genotype in BurstingStrategies-eve/_DataSubmission/DataSheetsAndCode/KineticsPlotStripes_Color/. *compiledResults_ectopicFlag.mat* and scripts to generate *Figure 3—figure supplement 3* are sorted by genotype in BurstingStrategies-eve/_DataSubmission/ DataSheetsAndCode/KineticsPlotsEndogenousEctopic/. Data to generate *Tables 2 and 3* is located in BurstingStrategies-eve/_DataSubmission/DataSheetsAndCode/BinStats/particle_counts/. Data sheets with detailed features of individual data points (identity and position of nuclei and MS2 foci; MS2 fluorescence; cpHMM-inference of fluorescence; cpHMM-inferred promoter state) are located in BurstingStrategies-eve/_DataSubmission/DataSheetsAndCode/BinStats/singleTraceFits/. Adobe Illustrator ai, eps, and png files for all figures are stored in BurstingStrategies-eve/_DataSubmission/ Figures/.

## Materials availability statement

Fly line expressing wild-type eveMS2-BAC is available at Bloomington *Drosophila* Stock center (stock #92368). Wild-type and mutant eveMS2-BAC constructs can be obtained through Vector Builder, as described in 'DNA constructs and fly lines', or by requesting them from the corresponding authors. For ChIP-seq data of enhancer associated-Zelda refer to 'Data Availability' section of *Harrison et al., 2011*. Details on sample preparation for confocal imaging are provided in the 'Embryo collection and mounting' section of *Berrocal et al., 2020*. Raw confocal imaging data, movies, and compiled particles files are stored in Zenodo, as described in 'Data and Code'. MATLAB code to segment MS2 signals in fruit fly embryos and generate compiled particles files, as described in 'Segmentation and quantification of movies', is publicly available in https://github.com/GarciaLab/mRNADynamics (*Reimer and Garcia, 2022*). For details on cpHMM scripts and methods refer to 'Data Availability' section in *Lammers et al., 2020*. DNA sequences, figures, and MATLAB code for data analysis are stored in https://github.com/aberrocal/BurstingStrategies-eve, as described in 'Data and Code'.

## Acknowledgements

We acknowledge Edward Pym and Michael Stadler for their insightful comments on the manuscript, as well as all the members of the Eisen Lab and the Garcia Lab for their valuable contributions to our discussions.

## Additional information

### Competing interests

Michael B Eisen: Michael B Eisen is former Editor-in-Chief of eLife. The other authors declare that no competing interests exist.

### Funding

| Funder | Grant reference number | Author |
| --- | --- | --- |
| Howard Hughes Medical Institute | | Michael B Eisen |
| NIH R01 Award | R01GM139913 | Hernan G Garcia |
| NIH Genomics and Computational Biology Training Grant | 5T32HG000047-18 | Nicholas C Lammers |

| Funder | Grant reference number | Author |
|---|---|---|
| Winkler Scholar Faculty Award in the Biological Sciences | | Hernan G Garcia |
| Koret-UC Berkeley-Tel Aviv University Initiative in Computational Biology and Bioinformatics | | Hernan G Garcia |
| Chan Zuckerberg Biohub-San Francisco Investigator | | Hernan G Garcia |
| University of California Institute for Mexico and the United States (UC MEXUS) Doctoral Fellowship | | Augusto Berrocal |
| NIH R01 Award | R01GM152815 | Hernan G Garcia |

The funders had no role in study design, data collection and interpretation, or the decision to submit the work for publication.

### Author contributions

Augusto Berrocal, Conceptualization, Data curation, Software, Formal analysis, Investigation, Visualization, Methodology, Writing – original draft, Writing – review and editing; Nicholas C Lammers, Data curation, Software, Formal analysis, Methodology; Hernan G Garcia, Conceptualization, Resources, Supervision, Funding acquisition, Investigation, Writing – original draft, Writing – review and editing; Michael B Eisen, Conceptualization, Resources, Supervision, Funding acquisition, Investigation, Methodology, Writing – original draft, Project administration

### Author ORCIDs

Augusto Berrocal (iD) https://orcid.org/0000-0002-6506-9071
Nicholas C Lammers (iD) https://orcid.org/0000-0001-6832-6152
Hernan G Garcia (iD) https://orcid.org/0000-0002-5212-3649
Michael B Eisen (iD) https://orcid.org/0000-0002-7528-738X

Reviewer #1 (Public Review): https://doi.org/10.7554/eLife.88671.3.sa1
Reviewer #2 (Public Review): https://doi.org/10.7554/eLife.88671.3.sa2
Reviewer #3 (Public Review): https://doi.org/10.7554/eLife.88671.3.sa3
Author response https://doi.org/10.7554/eLife.88671.3.sa4

# Additional files

### Supplementary files

- MDAR checklist

### Data availability

Raw data, Movies, and CompiledParticles files are stored in Zenodo (https://doi.org/10.5281/zenodo.7204096). Specific paths in this dataset are listed in Data and Code section of this article. MATLAB scripts and processed data for this analysis are stored in github (https://github.com/aberrocal/BurstingStrategies-eve, copy archived at *Berrocal et al., 2024*), as described in Data and Code section of this article.

The following dataset was generated:

| Author(s) | Year | Dataset title | Dataset URL | Database and Identifier |
|---|---|---|---|---|
| Berrocal A, Lammers NC, Garcia HG, Eisen MB | 2023 | Unified bursting strategies in ectopic and endogenous even-skipped expression patterns - Supplemental Data | https://doi.org/10.5281/zenodo.7204096 | Zenodo, 10.5281/zenodo.7204096 |

The following previously published dataset was used:

| Author(s) | Year | Dataset title | Dataset URL | Database and Identifier |
|---|---|---|---|---|
| Harrison MM, Kaplan T, Botchan MR, Eisen MB, Li XY | 2011 | Zelda binding in the early *Drosophila melanogaster* embryo marks regions subsequently activated at the maternal-to-zygotic transition | https://www.ncbi.nlm.nih.gov/geo/query/acc.cgi?acc=GSE30757 | NCBI Gene Expression Omnibus, GSE30757 |

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

## Appendix 1

### Supplemental Information

#### Complementary analysis of bursting parametersComplementary Analysis of Bursting Parameters

##### Bursting parameters in endogenous stripes controlled by mutant enhancers

Some stripes in this work are driven by mutant *eve* enhancers. We found that mutated enhancers modulate transcriptional output of endogenous stripes through the same mechanism as their wild-type counterparts: an increase in $k_{on}$ and $r$, while $k_{off}$ remains largely constant (***Figure 3—figure supplement 2***). In eveS1wt-eveS2Gt⁻ embryos (***Figure 3—figure supplement 2C***), *eve* stripe 2 is driven by a mutant *eve* stripe 2 enhancer. In eveS1Δ-eveS2wt embryos (***Figure 3—figure supplement 2D***), *eve* stripe 1 is active in the absence of *eve* stripe 1 enhancer, perhaps due to the activity of the late element. In eveS1Δ-eveS2Gt⁻ embryos (***Figure 3—figure supplement 2E***), *eve* stripe 2 is driven by a mutant *eve* stripe 2 enhancer and *eve* stripe 1 is active in the absence of *eve* stripe 1 enhancer. In all cases, our findings support the hypothesis that *eve*-regulatory elements employ a unified strategy to modulate transcriptional output. Bursting parameters of *eve* stripe 1 in embryos with a deleted *eve* stripe 1 enhancer (eveS1Δ-eveS2wt; eveS1Δ-eveS2Gt⁻) are of particular interest, as this expression is most likely activated by the *eve* late element. If this is the case, the *eve* late element would modulate transcriptional output through the same mechanism as the other enhancers, further underlining the unity of regulatory strategies across different *eve*-regulatory elements.

##### Comparison of bursting parameters between sets of nuclei grouped in endogenous and ectopic categories

We computed the bursting parameters of 3–6 bins per stripe (***Table 2***), depending on the amount of data obtained (see SI: ***Figure 3—figure supplement 1*** and Inference of bursting parameters in Materials and methods). To rule out the possibility that the observed $k_{on}$, $k_{off}$, and $r$ trends were skewed by the small number of bins, we aimed to redo our analysis with more data points per category (endogenous and ectopic), as a way to contrast bursting parameters between whole endogenous and ectopic regions and examine the bursting parameters trends that result from having 6–13 bins per category (***Table 3***).

We pooled together all nuclei from eveS1Δ-eveS2Gt⁻ embryos into endogenous (*eve* stripe 1, *eve* stripe 2, *eve* stripe 3, *eve* stripe 4) and ectopic sets (*eve* stripe 0, *eve* inter-stripe 1–2), and binned them by their mean MS2 fluorescence output to infer and compare their bursting parameters. We did the same analysis in wild-type, eveS1wt-eveS2Gt⁻, and eveS1Δ-eveS2wt embryos. We contrasted the bursting parameters of ectopic nuclei from eveS1Δ-eveS2Gt⁻ embryos against sets of endogenous nuclei from eveS1Δ-eveS2Gt⁻ eveS1wt-eveS2Gt⁻, eveS1Δ-eveS2wt, and wild-type embryos (***Figure 3—figure supplement 3***) and observed that all of them follow the same bursting strategy. Ectopic nuclei from eveS1Δ-eveS2Gt⁻ embryos boost transcriptional output through an increase in average $k_{on}$ (***Figure 3—figure supplement 3B***) and $r$ (***Figure 3—figure supplement 3D***), while $k_{off}$ remains largely the same, with only a minor drop at high mean MS2 fluorescence values (***Figure 3—figure supplement 3C***). The bursting parameters of endogenous nuclei from all the genotypes in this work follow the same trend.

