## [Editor Report · eLife assessment]

This manuscript is an **important** contribution toward understanding the mechanisms of transcriptional bursting. The evidence is considered **solid**. Questions regarding the broader advance, details of the analysis, and the models used in the analysis were addressed by the authors.

---

## [Referee Report · Reviewer #1 (Public Review)]

In this manuscript, the authors investigate whether enhancers use a common regulatory paradigm to modulate transcriptional bursting in both endogenous and ectopic domains using cis-regulatory mutant reporters of the eve transcriptional locus in early *Drosophila* embryogenesis.

The authors create a series of cis-regulatory BAC mutants of the eve stripe 1 and 2 enhancers by mutating the binding sites for the transcriptional repressor Giant in the stripe 2 minimal response element (MRE) independently or in combination with deletion of the stripe 1 enhancer sequence. With these enhancer mutations, they are able to generate conditions in which eve is ectopically expressed. Next, the authors investigate if nuclei in these "ectopic" regions have similar transcriptional kinetics to the "endogenous"-expressing eve+ nuclei. They show that bursting parameters are unchanged when comparing endogenous and ectopic gene expression regions. Under a scheme of a 2-state model, the eveS1Δ-EveS2Gt- reporter modulates transcription by increasing the active state switching rate (kon) and the initiation rate (r) while maintaining a constant inactive state switching rate.

Based on these results, the authors support a model whereby kinetic regimes are encoded in the cis-regulatory sequences of a gene instead of imposed by an evolving trans-regulatory environment.

The question asked in this manuscript is important and the eve locus represents an ideal paradigm to address it in a quantitative manner. Most of the results are correctly interpreted and well-presented.

---

## [Referee Report · Reviewer #2 (Public Review)]

The manuscript by Berrocal et al. asks if shared bursting kinetics, as observed for various developmental genes in animals, hint towards a shared molecular mechanism or result from natural selection favoring such a strategy. Transcription happens in bursts. While transcriptional output can be modulated by altering various properties of bursting, certain strategies are observed more widely. As the authors noted, recent experimental studies have found that even-skipped enhancers control transcriptional output by changing burst frequency and amplitude while burst duration remains largely constant. The authors compared the kinetics of transcriptional bursting between endogenous and ectopic gene expression patterns. It is argued that since enhancers act under different regulatory inputs in ectopically expressed genes, adaptation would lead to diverse bursting strategies as compared to endogenous gene expression patterns. To achieve this goal, the authors generated ectopic even-skipped transcription patterns in fruit fly embryos. The key finding is that bursting strategies are similar in endogenous and ectopic even-skipped expression. According to the authors, the findings favor the presence of a unified molecular mechanism shaping even-skipped bursting strategies. This is an important piece of work. Everything has been carried out in a systematic fashion.

---

## [Referee Report · Reviewer #3 (Public Review)]

In this manuscript by Berrocal and coworkers, the authors do a deep dive into the transcriptional regulation of the eve gene in both an endogenous and ectopic background. The idea is that by looking at eve expression under non-native conditions, one might infer how enhancers control transcriptional bursting. The main conclusion is that eve enhancers have not evolved to have specific behaviors in the eve stripes, but rather the same rates in the telegraph model are utilized as control rates even under ectopic or 'de novo' conditions. For example, they achieve ectopic expression (outside of the canonical eve stripes) through a BAC construct where the binding sites for the TF Giant are disrupted along with one of the eve enhancers. Perhaps the most general conclusion is that burst duration is largely constant throughout at ~ 1 - 2 min. This conclusion is consistent with work in human cell lines that enhancers mostly control frequency and that burst duration is largely conserved across genes, pointing to an underlying mechanistic basis that has yet to be determined.

---

## [Author Response]

The following is the authors’ response to the original reviews.

**Reviewer #1 (Public Review):**
[...] Based on these results, the authors support a model whereby kinetic regimes are encoded in the cis-regulatory sequences of a gene instead of imposed by an evolving trans-regulatory environment.The question asked in this manuscript is important and the eve locus represents an ideal paradigm to address it in a quantitative manner. Most of the results are correctly interpreted and well-presented. However, the main conclusion pointing towards a potential "unified theory" of burst regulation during *Drosophila* embryogenesis should be nuanced or cross-validated.

Our results and those of others suggest that different developmental genes follow unified—yet different—transcriptional control strategies whereby different combinations of bursting parameters are regulated to modulate gene expression: burst frequency and amplitude for eve (Berrocal et al., 2020), and burst frequency and duration for gap genes (Zoller et al., 2018). In light of the aforementioned works, we can only claim that our results suggest a unified strategy for eve, our case of study, as we observe that eve regulatory strategies are robust to disruption of enhancers and binding sites. In the Discussion section of our revised manuscript, we will emphasize that the bursting control strategy we uncovered for eve does not necessarily apply to other genes, and speculate in more detail that genes that employ the same strategy of transcriptional bursting may be grouped in families that share a common molecular mechanism of transcription.

Manuscript updates:

We have emphasized in the Discussion section that our claim of unified strategies pertains exclusively to the bursting behavior of the gene even-skipped, and do not necessarily extend to other genes. To clarify this point, we referenced the findings of (Zoller, Little, and Gregor 2018) and (Chen et al. 2023), who observed that the bursting control strategy of *Drosophila* gap genes relies on the modulation of burst frequency and duration. Additionally, we cited the findings of (Syed, Duan, and Lim 2023), who reported a decrease in bursting amplitude and duration upon disruption of Dorsal binding sites on the snail minimal distal enhancer. Both examples describe bursting control strategies that differ from the modulation of burst frequency and amplitude observed for even-skipped.

In addition to the lack of novelty (some results concerning the fact that koff does not change along the A/P axis/the idea of a 'unified regime' were already obtained in Berrocal et al 2020),...

Unfortunately, we believe there is a misunderstanding in terms of what we construe as novelty in our work. In our previous work (Berrocal et al., 2020), we observed that the seven stripes of even-skipped (eve) expression modulate transcriptional bursting through the same strategy—bursting frequency and amplitude are controlled to yield various levels of mRNA synthesis, while burst duration remains constant. We reproduce that result in our paper, and do not claim any novelty. However, what was unclear is whether the observed eve bursting control strategy would only exist in the wild-type stripes, whose expression—we reasoned—is under strong selection due to the dramatic phenotypic consequences of eve transcription, or if eve transcriptional bursting would follow the same strategy under trans-regulatory environments that are not under selection to deliver specific spatiotemporal dynamics of eve expression. Our results—and here lies the novelty of our work—support the second scenario, and point to a model where eve bursting strategies do not result from adaptation of eve activity to specific trans-regulatory environments. Instead, we speculate that a molecular mechanism constrains eve bursting strategy whenever and wherever the gene is active. This is something that we could not have known from our first study in (Berrocal et al., 2020) and constitutes the main novelty of our paper. To put this in other words, the novelty of our work does not rest on the fact that both burst frequency and amplitude are modulated in the endogenous eve pattern, but that this modulation remains quantitatively indistinguishable when we focus on ectopic areas of expression. We will make this point clearer in the Introduction and Discussion section of our revised manuscript.

Manuscript updates:

We have clarified this point in both the Introduction and Discussion sections. In the updated Introduction, we state that while our previous work (Berrocal et al. 2020) examined bursting strategies in endogenous expression regions that are, in principle, subject to selection, the present study induced the formation of ectopic expression patterns to probe bursting strategies in regions presumably devoid of evolutionary pressures. In the Discussion section, we highlight that the novelty of our work lies in the insights derived from the comparative analysis between ectopic and endogenous regions of even-skipped expression, an aspect not addressed in our previous work.

… note i the limited manipulation of TF environment;...

We acknowledge that additional genetic manipulations would make it possible to further test the model. However, we hope that the reviewer will agree with us that the manipulations that we did perform are sufficient to provide evidence for common bursting strategies under the diverse trans-regulatory environments present in wild-type and ectopic regions of gene expression. In the Discussion section of our revised manuscript, we will elaborate further on the kind of genetic manipulations (e.g., probing transcriptional strategies that result from swapping promoters in the context of eve-MS2 BAC; or quantifying the impact on eve transcriptional control after performing optogenetic perturbations of transcription factors and/or chromatin remodelers) that could shed further light on the currently undefined molecular mechanism that constrains eve bursting strategies, as a mean to motivate future work.

Manuscript updates:

In our Discussion section, we elaborated on proposed manipulations of the transcription factor environment to elucidate the molecular mechanisms behind even-skipped bursting control strategies. We began by listing studies linking transcription factor concentration to bursting control strategies, such as (Hoppe et al. 2020), who observed that the natural BMP (Bone Morphogenetic Protein) gradient shapes bursting frequency of target genes in *Drosophila* embryos. And (Zhao et al. 2023), who used the LEXY optogenetic system to modulate Knirps nuclear concentration and observed that this repressor acts on eve stripe 4+6 enhancer by gradually decreasing bursting frequency until the locus adopts a reversible quiescent state. Then, we proposed performing systematic LEXY-mediated modulation of critical transcription factors (Bicoid, Hunchback, Giant, Kruppel, Zelda) to understand the extent of their contribution to the unified even-skipped bursting strategies.

To better frame the hypothesis that the even-skipped promoter defines strategies of bursting control, we added a reference to the work of (Tunnacliffe, Corrigan, and Chubb 2018). This study surveyed 17 actin genes with identical sequences but distinct promoters in the amoeba *Dictyostelium discoideum*, and found that all genes display different bursting strategies. Their findings, together with the previously cited work by (Pimmett et al. 2021) and (Yokoshi et al. 2022), suggest a critical role of gene promoters in constraining the bursting strategies of eukaryotic genes.

… ii the simplicity with which bursting is analyzed (only a two-state model is considered, and not cross-validated with an alternative approach than cpHMM) and…

Based on our previous work (Lammers et al., 2020), and as described in the SI Section of the current manuscript: Inference of Bursting Parameters, we selected a three-state model (OFF, ON1, ON2) under the following rationale: transcription of even-skipped in pre-gastrulating embryos occurs after DNA replication, and promoters on both sister chromatids remain paired. Most of the time these paired loci cannot be resolved independently using conventional microscopy. As a result, when we image an MS2 spot, we are actually measuring the transcriptional dynamics of two promoters. Thus, each MS2-fluorescent spot may result from none (OFF), one (ON1) or two (ON2) sister promoters being in the active state. Following our previous work, we analyzed our data assuming the three-state model (OFF, ON1, ON2), and then, for ease of presentation, aggregated ON1 and ON2 into an effective single ON state. As for the lack of an alternative model, we chose the simplest model compatible with our data and our current understanding of transcription at the eve locus. With this in mind, we do not rule out the possibility that more complex processes—that are not captured by our model—shape MS2 fluorescence signals. For example, promoters may display more than two states of activity. However, as shown in (Lammers et al., 2020 - SI Section: G. cpHMM inference sensitivities), model selection schemes and cross-validation do not give consistent results on which model is more favorable; and for the time being, there is not a readily available alternative to HMM for inference of promoter states from MS2 signal. For example, orthogonal approaches to quantify transcriptional bursting, such as smFISH, are largely blind to temporal dynamics. As a result, we choose to entertain the simplest two-state model for each sister promoter. We appreciate these observations, as they point out the need of devoting a section in the supplemental material of our revised manuscript to clarify the motivations behind model selection.

Manuscript updates:

We have devoted the new Supplemental Material section “Selection of a three-state model of promoter activity and a compound Hidden Markov Model for inference of promoter states from MS2 fluorescent signal” to clarify the rationale behind our selection of a three-state promoter activity model. Since transcription in pre-gastrulating *Drosophila* embryos occurs after DNA replication, each MS2-active locus contains two unresolvable sister promoters that can either be inactive (OFF), one active (ON1), or both active (ON2).

Next, we elaborated on the conversion of a three-state model into an effective two-state model for ease of presentation and described how the effective two-state model parameters—kon (burst frequency), koff-1 (burst duration), and r (burst amplitude)—were calculated.

Additionally, we acknowledged that while the three-state model of promoter activity is the simplest model compatible with our current understanding of transcription in the even-skipped locus, we do not rule out the possibility that even-skipped transcription may be described by more complex models that include multiple states beyond ON and OFF. Finally, we referenced (Lammers et al. 2020) who asserted that while all inferences of promoter states computed from confocal microscopy of MS2/PP7 fluorescence data rely on Hidden Markov models, cross-comparisons between one, two, or multiple-state Hidden Markov models do not yield consistent results regarding which is more accurate. We close the new section by proposing that state-of-the-art microscopy and deconvolution algorithms to improve signal-to-noise-ratio may offer alternatives to the inference of promoter states.

… (iii) the lack of comparisons with published work.

We thank the reviewer for pointing this out. In the current discussion of our manuscript, we compare our findings to recent articles that have addressed the question of the origin of bursting control strategies in *Drosophila* embryos (Pimmett et al., 2021; Yokoshi et al., 2022; Zoller et al., 2018). Nevertheless, we acknowledge that we failed to include references that are relevant to our study. Thus, our revised Discussion section must include recent results by (Syed et al., 2023), which showed that the disruption of Dorsal binding sites on the snail minimal distal enhancer results in decreased amplitude and duration of transcription bursts in fruit fly embryos. Additionally, we have to incorporate the study by (Hoppe et al., 2020), which reported that the *Drosophila* bone morphogenetic protein (BMP) gradient modulates the bursting frequency of BMP target genes. References to thorough studies of bursting control in other organisms, like *Dictyostelium discoideum* (Tunnacliffe et al., 2018), are due as well.

Manuscript updates:

As mentioned in the updates above, our revised manuscript now includes long due references to studies by (Syed, Duan, and Lim 2023), (Hoppe et al. 2020), (Tunnacliffe, Corrigan, and Chubb 2018), and (Chen et al. 2023). All of which are relevant for our current workk.

**Reviewer #2 (Public Review):**
The manuscript by Berrocal et al. asks if shared bursting kinetics, as observed for various developmental genes in animals, hint towards a shared molecular mechanism or result from natural selection favoring such a strategy. Transcription happens in bursts. While transcriptional output can be modulated by altering various properties of bursting, certain strategies are observed more widely. As the authors noted, recent experimental studies have found that even-skipped enhancers control transcriptional output by changing burst frequency and amplitude while burst duration remains largely constant. The authors compared the kinetics of transcriptional bursting between endogenous and ectopic gene expression patterns. It is argued that since enhancers act under different regulatory inputs in ectopically expressed genes, adaptation would lead to diverse bursting strategies as compared to endogenous gene expression patterns. To achieve this goal, the authors generated ectopic even-skipped transcription patterns in fruit fly embryos. The key finding is that bursting strategies are similar in endogenous and ectopic even-skipped expression. According to the authors, the findings favor the presence of a unified molecular mechanism shaping even-skipped bursting strategies. This is an important piece of work. Everything has been carried out in a systematic fashion. However, the key argument of the paper is not entirely convincing.

We thank the reviewer, as these comments will enable us to improve the Discussion section and overall logic of our revised manuscript. We agree that the evidence provided in this work, while systematic and carefully analyzed, cannot conclusively rule out either of the two proposed models, but just provide evidence supporting the hypothesis for a specific molecular mechanism constraining eve bursting strategies. Our experimental evidence points to valuable insights about the mechanism of eve bursting control. For instance, had we observed quantitative differences in bursting strategies between ectopic and endogenous eve domains, we would have rejected the hypothesis that a common molecular mechanism constrains eve transcriptional bursting to the observed bursting control strategy of frequency and amplitude modulation. Thus, we consider that our proposition of a common molecular mechanism underlying unified eve bursting strategies despite changing trans-regulatory environments is more solid. On the other hand, while our model suggests that this undefined bursting control strategy is not subject to selection acting on specific trans-regulatory environments, it is not trivial to completely discard selection for specific bursting control strategies given our current lack of understanding of the molecular mechanisms that shape the aforesaid strategies. Indeed, we cannot rule out the hypothesis that the observed strategies are most optimal for the expression of eve endogenous stripes according to natural selection, and that these control strategies persist in ectopic regions as an evolutionary neutral “passenger phenotype” that does not impact fitness. We recognize the need to acknowledge this last hypothesis in the updated Introduction and Discussion sections of our manuscript. Further studies will be needed to determine the mechanistic and molecular basis of eve bursting strategies.

Manuscript updates:

In this work, we compared strategies of bursting control between endogenous and ectopic regions of even-skipped expression. Different strategies between both regions would suggest that selective pressure maintains defined bursting strategies in endogenous regions. Conversely, similar strategies in both ectopic and endogenous regions would imply that a shared molecular mechanism constrains bursting parameters despite changing trans-regulatory environments.

In our updated Discussion section, we acknowledge that while our work provides evidence supporting the second hypothesis, we cannot conclusively rule out the possibility that the observed strategies were selected as the most optimal for endogenous even-skipped expression regions and that ectopic regions retain such optimal bursting strategies as an evolutionary neutral “passenger phenotype”.

**Reviewer #3 (Public Review):**
In this manuscript by Berrocal and coworkers, the authors do a deep dive into the transcriptional regulation of the eve gene in both an endogenous and ectopic background. The idea is that by looking at eve expression under non-native conditions, one might infer how enhancers control transcriptional bursting. The main conclusion is that eve enhancers have not evolved to have specific behaviors in the eve stripes, but rather the same rates in the telegraph model are utilized as control rates even under ectopic or 'de novo' conditions. For example, they achieve ectopic expression (outside of the canonical eve stripes) through a BAC construct where the binding sites for the TF Giant are disrupted along with one of the eve enhancers. Perhaps the most general conclusion is that burst duration is largely constant throughout at ~ 1 - 2 min. This conclusion is consistent with work in human cell lines that enhancers mostly control frequency and that burst duration is largely conserved across genes, pointing to an underlying mechanistic basis that has yet to be determined.

We thank the reviewer for the assessment of our work. Indeed, evidence from different groups (Berrocal et al., 2020; Fukaya et al., 2016; Hoppe et al., 2020; Pimmett et al., 2021; Senecal et al., 2014; Syed et al., 2023; Tunnacliffe et al., 2018; Yokoshi et al., 2022; Zoller et al., 2018) is coming together to uncover commonalities, discrepancies, and rules that constrain transcriptional bursting in *Drosophila* and other organisms.

Additional updates to the manuscript

(1) In our current study, we observed the appearance of a mutant stripe of even-skipped expression beyond the anterior edge of eve stripe 1, which we refer to as eve stripe 0. This stripe appeared in embryos with a disrupted eve stripe 1 enhancer. In a previous study, (Small, Blair, and Levine 1992) reported a “head patch” of even-skipped expression while assaying the regulation of reporter constructs carrying the minimal regulatory element of eve stripe 2 enhancer alone. In our updated manuscript, we state that it is tempting to identify our eve stripe 0 with the previously reported head patch. (Small, Blair, and Levine 1992) speculated that this head patch of even-skipped expression appeared as a result of regulatory sequences present in the P-transposon system they used for genomic insertions. However, P-transposon sequences are not present in our experimental design. Thus, the appearance of eve stripe 0 indicates a repressive role of the eve stripe 1 enhancer at the anterior end of the embryo and may imply that the minimal regulatory element of the eve stripe 2 enhancer, as probed by (Small, Blair, and Levine 1992), can drive the expression of the head patch/eve stripe 0 when the eve stripe 1 enhancer is not present.

(2) In our current analysis, we observed that the disruption of Gt-binding sites on the eve stripe 2 enhancer synergizes with the deletion of the eve stripe 1 enhancer, as double mutant embryos display more ectopic expression in their anterior regions than embryos with only disrupted Gt-binding sites. While this may indicate that the repressive activity of eve stripe 1 enhancer synergizes with the repression exerted by Giant, other unidentified transcription factors may be involved in this repressive synergy. In the updated manuscript we clarified that unidentified transcription factors may bind in the vicinity of Gt-binding sites. The hypothesis that Gt-binding sites recognize other transcription factors was proposed by (Small, Blair, and Levine 1992), as they observed that the anterior expansion of eve stripe 2 resulting from Gt-binding site deletions was “somewhat more severe” than expansion observed in embryos carrying null-Giant alleles.